# Legal and Regulatory Approaches to Rehabilitation Planning: A Concise Overview of Current Laws and Policies Addressing Access to Rehabilitation in Five European Countries

**DOI:** 10.3390/ijerph17124363

**Published:** 2020-06-11

**Authors:** Aditi Garg, Dimitrios Skempes, Jerome Bickenbach

**Affiliations:** 1Department of Health Sciences and Medicine, University of Lucerne, Frohburgstrasse 3, 6002 Luzern, Switzerland; dimitrios.skempes@paraplegie.ch (D.S.); jerome.bickenbach@paraplegie.ch (J.B.); 2Swiss Paraplegic Research, Guido A. Zäch Str. 4, 6207 Nottwil, Switzerland

**Keywords:** rehabilitation, health policy, health planning, health services for disabled persons, Convention on the Rights of Persons with Disabilities, Europe, health laws, health legislation, health strategy

## Abstract

*Background*: The rising prevalence of disability due to noncommunicable diseases and the aging process in tandem with under-prioritization and underdevelopment of rehabilitation services remains a significant concern for European public health. Over recent years, health system responses to population health needs, including rehabilitation needs, have been increasingly acknowledging the power of law and formal written policies as strategic governance tools to improve population health outcomes. However, the contents and scope of enacted legislation and adopted policies concerning rehabilitation services in Europe has not been synthesized. This paper presents a concise overview of laws and policies addressing rehabilitation in five European countries. *Methods*: Publicly available laws, policies, and national action plans addressing rehabilitation issues of Sweden, Italy, Germany, the Netherlands and the United Kingdom were reviewed and descriptive documents analyzed. Actions found in national health policies were also evaluated for compliance with the key recommendations specified in the World Health Organization’s Rehabilitation 2030: Call for Action. *Results:* Across countries, legal and policy approaches to rehabilitation planning varied in scope and reach. While all countries entitle citizens to rehabilitation services, comprehensiveness of coverage varied. Health legislation of Germany and Netherlands recognizes access to rehabilitation as a human right for persons with disabilities, while Sweden and the United Kingdom acknowledge its importance in disability laws for achieving substantive equality for persons with disabilities. Regarding policies, in all countries but Italy, targeted universalism remains the predominant strategy governing rehabilitation services, as demonstrated by the lack of comprehensive, national action plans for rehabilitation addressing the general population. Nevertheless, references found in disease specific policies indicate a solid consensus that rehabilitation remains an integral component of the care continuum for those experiencing disability. *Conclusion*: Although a universal approach to rehabilitation coverage is institutionalized in national legislation of the countries examined, this approach is not expressed in formal policies. Targeted strategies aiming to ensure access to subpopulation groups with higher perceived needs for rehabilitation prevail, indicating a strong political will towards the reduction of health inequalities and the promotion of human rights of people experiencing disability. Results obtained from conducting this descriptive review provide the basis for future appraisals of the situation regarding rehabilitation service and policy development in Europe.

## 1. Introduction

The World Health Organization (WHO) estimates that about 15% of the world’s population experiences some degree of disability, of whom 2–4% (around 110–190 million persons) experience significant difficulties in functioning [1]. The global prevalence of disability is increasing due to, among others, population aging and the rapid increase of chronic noncommunicable diseases (NCDs), with major implications for individuals and health systems. At the health system level, the increasing number of people experiencing disability, as well as those living with comorbidities, means there is a rising demand for healthcare and rehabilitation services, adding pressure to already constrained health and household budgets [2]. 

Defined broadly as “a set of measures that assist individuals who experience, or are likely to experience, disability to achieve and maintain optimal functioning in interaction with their environments” [1] (p. 96), rehabilitation is a key strategy for achieving population health and well-being as it promotes recovery from illness, improves human functioning, and maximizes opportunities for social participation [3]. It addresses problems at the impairment level as well as environmental factors that have an impact on functioning. Rehabilitation encompasses a wide range of medical interventions and specific measures required by persons experiencing disability, from diagnosis and therapy to assistive technology and psychosocial support [1,4]. As a person-centered strategy, rehabilitation enhances personal autonomy and empowers people to take full control over their lives [4]. Moreover, rehabilitation has the potential to improve efficiency and reduce healthcare costs by reducing secondary complications associated with primary health conditions and the subsequent utilization of expensive acute services [5]. Also, rehabilitation is associated with reductions in hospital length of stay for various patient groups, as well as increased labor market participation for both individual patients and their carers. Importantly, the Convention on the Rights for Persons with Disabilities (CRPD) [6] recognized access to rehabilitation as a human right, highlighting the obligation of States to ensure equitable access to appropriate rehabilitation services and supports for all persons with disabilities [7].

In Europe, as in other parts of the world, despite progress in dealing with chronic disabling diseases [8,9,10], notably with regard to prevention and control of NCDs [11], the adequate provision of rehabilitation services to all those who need them, especially including persons with disabilities [12], remains a difficult challenge [13]. Recognizing that there is an urgent need to address the many issues surrounding the development of rehabilitation services, the WHO convened a stakeholder’s group to determine the direction that policy decision makers should take. The Rehabilitation 2030: A Call for Action [14] acknowledged that strengthening health systems to provide rehabilitation services is crucial to progressively realize universal health coverage (UHC), and specifically recommends that countries (a) create strong leadership and political support for rehabilitation, (b) strengthen rehabilitation planning, and, most importantly, (c) incorporate rehabilitation in UHC [15].

Clearly, WHO’s call for action is an opportunity for all stakeholders to accelerate their efforts to integrate rehabilitation in national health planning and programming so that patients have access to rehabilitation appropriate to their needs. For such efforts to be successful, there is a need for baseline information on the extent to which health systems currently incorporate rehabilitation into national laws and regulations, particularly those governing health insurance coverage, as well as in specific policies and programs of action that address priority health issues. The provision of rehabilitation is recognized as a critical issue on European health and social agendas, such as the recent European policy framework entitled Health 2020 [16] and the action plan for the implementation of the European strategy on the prevention and control of noncommunicable diseases [17]. Despite rehabilitation being acknowledged by policymakers and administrators as crucial for improved access to health care, it is unclear if and to what extent rehabilitation is included in national UHC frameworks, and how policies guide the delivery of rehabilitation services in health systems. There are very few detailed examinations of the legal and policy environment of rehabilitation. This study sought to identify, analyze, and compare health laws and policies in five European Union countries to better understand the current landscape of rehabilitation governance and identify gaps in UHC frameworks and policies. The study addressed the following questions:1How do national laws govern the provision of rehabilitation?2What aspects of rehabilitation services are subject to State regulation?3How is rehabilitation being addressed in national health policies and strategies?4What are the common (or divergent) approaches to rehabilitation policy development?

## 2. Methods

We conducted a literature review to identify national health legislation and policies to examine how selected European Union (EU) countries have addressed rehabilitation in health planning. Evidence generated from the review and analysis of textual data was used to create a summary report for each country and a narrative synthesis describing key findings and discussing implication for European health policy. 

### 2.1. Country Selection

The following five countries were examined in this study: the United Kingdom (UK), Germany (DE), Sweden (SW), Italy (IT), and the Netherlands (NL). The countries were deliberately chosen for diversity (Table 1). They range from those with national health care systems and strong tradition in developing policies and programs for the societal inclusion of disabled persons (UK, Sweden) to those with statutory health insurance systems and strong rehabilitation services (Germany, Netherlands), are from several subregions of the EU region, and have developed their rehabilitation services at different levels. The countries also reflect different historical backgrounds (e.g., the post-World War II era for Germany and Italy). These countries were also selected for how their commitment to and their readiness to rehabilitation lends to exploration of the key research questions of the study.

### 2.2. Design

A flexible scoping review and narrative synthesis methodology was adopted to examine the positioning of rehabilitation in health laws and rehabilitation service development in key national health strategies and action plans. Data sources and terminology used in the search strategy were specified for identifying, selecting, and analyzing national health laws and policies according to set inclusion and exclusion criteria.

#### 2.2.1. Data Sources

The evidence gathered from the review was mostly found in the form of grey literature, including enacted legislations, policy documents, strategic plans, and action plans. Key sources included scientific databases, websites of governmental and nongovernmental organizations, and publicly available generic search engines through which hand-searching was performed. The primary database used to find data regarding rehabilitation coverage in national legislation was N-Lex. Specific data sources in regard to searching for policy documents included health system and country reports published by the Commonwealth Fund, the European Observatory on Health System and Policies, States’ reports to the Committee on the Rights of Persons with Disabilities, and other supporting documents to guide the identification of policies. Most of the information found in national policies and action plans relied on general Google and Google Scholar searches.

#### 2.2.2. Terminology and Search Terms

Prior to performing the literature search, search terms were clearly defined according to set eligibility criteria with respect to rehabilitation and health systems.

(1) Rehabilitation

To date, there is no widely agreed definition for rehabilitation. In the realm of health, rehabilitation is often used to denote the provision of medical treatment programs and interventions aiming to restore functioning. In this study, rehabilitation is defined according to WHO as “a set of interventions designed to optimize functioning and reduce disability in individuals with health conditions in interaction with their environment” [18] (p. 1). This definition considers an expanded group of beneficiaries of rehabilitation beyond the class of disabled people, including those with health conditions who experience disability, such as people diagnosed with early stage cancer, diabetes, or depression. According to this definition, rehabilitation can be seen as a collection of interventions including medical interventions that may be required at the acute phase of diagnosis to prevent the loss of function associated with health conditions. In addition, it incorporates the International Classification of Functioning’s (ICF) interactional model of disability which recognizes that disability may be experienced and expressed differently in different individuals. For this reason, rehabilitation responses should be centered around individual-specific needs, preferences, and values. Overall, WHO’s notion of rehabilitation reflects a view of rehabilitation as a strategy for improving population health and functioning across the care continuum and lifespan. In practical terms, rehabilitation covers a large and diversified range of health services intended to improve physical, mental, cognitive, and sensory abilities in functioning. In most countries, these services are linked to national health care programs and the public health system, and are important resources for policy implementation in the field of health and disability, as they seek to optimize capacity for individuals for full participation in society [19]. Typically, the health sector is responsible for regulating access and provision for rehabilitative health services, but does not necessarily have to be the provider of them [19]. In clinical terms, rehabilitation is a time-limited process that is implemented from the acute phase to the post-acute phase. It is a goal-oriented process that involves the identification of a person’s problems and needs, defining desired goals, planning and implementing interventions, and assessing their effects. The outcomes of rehabilitation are usually achieved through the single or combined application of principles and techniques of rehabilitation medicine, physiotherapy, occupational therapy, speech and language therapy, and provision of assisted products [19]. Other supportive interventions such as psychosocial counseling, caregiver training, installation of assistive equipment, as well as measures focused on return to work may indirectly promote and improve an improved individual health and functioning [19]. These are all important ingredients of rehabilitation that should be considered in any review assessing legislative parameters, policies, and programmatic initiatives aiming to improve or strengthen rehabilitation at the national level.

Relevant terms used to capture rehabilitation-related laws and policies included “rehabilitation”, “rehabilitation medicine”, “physical therapy”, “occupational therapy”, “speech language therapy”, “assistive devices”, and “persons with disabilities”. 

(2) Health Legislation and Policies 

According to the WHO, health law is the “area of law concerned with the health of individuals and populations, the provision of health care and the operation of the health system” [20]; more specifically, these “[binding] rules make up the legal framework, or legal architecture for health” [21]. This health legislation may encompass both primary (statutes, acts) and secondary (e.g., decrees, bylaws, and court precedents) legislation [22].

Policies and strategies were terms commonly used interchangeably in health policy research. WHO refers to health policy as “decisions, plans, and actions that are undertaken to achieve specific health care goals within a society” [23]. It further defines policy as “a set of decisions or commitments to pursue courses of action aimed at achieving defined goals for improving health, stating or inferring the values that underpin these decisions” [24]. A strategy is defined by the WHO as “a series of broad lines of action intended to achieve a set of goals and targets set out within a policy or programme” [24].

Terms were used in combination with the rehabilitation terms specified above to facilitate the identification of national health legislation and policies with respect to rehabilitation care. For laws, search terms included “health law”, “health legislation”, “health insurance law”, and “health service framework”. Search terms used to retrieve policy documents included “[national] program strategy”, “[national] strategic plan”, and “national health strategy”. Additional search terms used within legislation and policy documents, in various combinations with rehabilitation terms, included “access to health services”, “breadth of integration”, “comprehensive health services”, “comprehensiveness of care”, “coverage”, “health insurance”, “health planning”, “health service”, “national disease”, “personal health services”, “health law”, “health legislation”, and “primary health care”.

#### 2.2.3. Document Identification

Criteria for the inclusion and exclusion of documents, outlined in Table 2, were created to efficiently identify documents and to facilitate selection and analysis of coverage of rehabilitation services in health laws and policies. Each search term outlined in Section 2.2.2 was translated into appropriate languages in Table 2 to perform searches in the selected countries (Section 2.1). 

In general, both legislation and policy documents were excluded if published in a language other than English or the national language of the country examined, published outside of the specified time frame (Table 2), or clearly unrelated to the health sector.

The inclusion criteria for health legislation required that health service coverage be a substantial component of the screened documents, including descriptions of the essential health benefits package outlined in health laws, with an explicit focus on the development and provision of rehabilitation services and regulation of other important aspects of rehabilitation service delivery.

In terms of searching literature for rehabilitation coverage in policies, publicly available national health policies, action plans, and strategies were examined. Particularly, disability- and rehabilitation-specific national health policy documents were studied to assess how and to what degree rehabilitation is positioned in national health planning and among key health priorities. Different policies searched for were distinguished, including global policies, aging policies, and other policies from various health programmatic areas. Only policy documents at the national level were searched for, in addition to agency policies with implications for rehabilitation.

#### 2.2.4. Document Selection and Analysis

After applying inclusion and exclusion criteria to the search strategy, evidence regarding breadth and depth of health service coverage was found primarily in health legislation and strategy and policy documents. Documents found according to inclusion and exclusion criteria were verified by a second reviewer and discrepancies were resolved through discussion and consensus with a third reviewer. The analysis of regulations, such as laws regulating the rehabilitation workforce, was beyond the scope of this study. For this reason, national health laws and policies were analyzed, and regulations with respect to rehabilitation were not. As expected, limited literature was found in scientific databases, and few peer-reviewed articles pertained to rehabilitation and health services coverage in data sources, including Pubmed/Medline. 

To facilitate analysis of data retrieved, a data extraction sheet was developed to extract and collate information from health laws and legislation on the breadth and depth of coverage for rehabilitation care, types of services, reimbursement mechanisms, health financing, human resources, and health insurance into succinct narratives describing the legal context of rehabilitation in each country. Concise metasummaries of textual data were also developed to facilitate collation of relevant information and presentation of findings. An extraction table was created to retrieve action statements and key action areas from selected policy and strategy documents. Subsequently, a comparative analysis assessed the alignment of action statements and key action areas with the WHO Rehabilitation Call for Action report, in terms of rehabilitation care. 

The raw data of health legislations and policy documents retrieved are available online through Appendix A. Figure 1 displays a flow chart summarizing the review process. In total, there were 115 results identified and screened through the search, and a total of 83 documents were included in the final analysis. 

## 3. Results

This section provides a detailed description of the inclusion of rehabilitation services in national laws (*n* = 54) and policies (*n* = 29) that govern the provision of health care in the United Kingdom, Netherlands, Sweden, Germany, and Italy.

Figure 2 below visually highlights the main findings gathered from the tabulated data in Table A1 and Table A2.

Overall, similar trends regarding the direct mention of rehabilitation coverage or service provisions were found for both health policy instruments and legal documents retrieved in this review. There are more references to rehabilitation in policy documents than in legislation documents. Nearly half of reviewed policies and action plans partially addressed rehabilitation, and nearly one third of gathered documents directly addressed rehabilitation in both legislation and policy.

Sectoral policies and laws—those administered by the nation’s ministry of health—comprised the majority of documents reviewed and followed similar distributions to those of total documents with respect to rehabilitation coverage. Few cross-sectoral policies were assessed in this review. The majority of cross-sectoral laws—those primarily interfacing health and disability human rights legislation—did not address rehabilitation care.

Consideration of access to rehabilitation services varied across health system and disease-specific programming areas. Rehabilitation was addressed most frequently at the primary care level, followed by conditions of mental health and chronic disability. For certain prevalent health conditions, such as cancer and cardiopulmonary diseases, rehabilitation appears to be a neglected aspect of health planning strategies; only two of twenty-nine policies were specific to these conditions. Surprisingly, we found rehabilitation care was not addressed in disease-specific plans, such as diabetes, despite promising evidence in improving health outcomes for this condition [26,27]. Overall, rehabilitation is addressed as a key component of the care continuum, but it does not seem to be addressed at the system level.

Regarding areas of legislation, the majority of documents concerned general healthcare legislation, legislation on the design of the health insurance benefits package, and patient and disability rights legislation. More specifically, concerning legislation governing general health care issues, nearly half of the laws did not address rehabilitation care (44%). Similar trends exist across other areas, including health insurance and benefit legislations, as well as legislation for patient and disability rights. Notably, only five of the total fifty-four health legislation documents examined focused specifically on rehabilitation coverage. 

In the following text, we present a country-by-country analysis with a more descriptive review of the information retrieved regarding rehabilitation care and provision in the countries selected for this study.

### 3.1. United Kingdom

The UK’s health care system is primarily publicly financed by the National Health Service (NHS), largely through revenues from general taxation, and is mostly free at the point of access [28]. Health care in the UK is decentralized and, with the exception of England, Wales, Northern Ireland, and Scotland, independently make decisions regarding the organization of health services [28]. Primary goals include finding ways to improve integration of health and social care, increasing cost effectiveness and efficiency of the health care system, and providing patients with higher quality health care services [28].

In recent years, less emphasis has been placed on development of rehabilitation and intermediate care in comparison to inpatient hospital care [28]. 

Many of the UK laws either ignore or only indirectly address rehabilitation service provision. A more common theme in the UK laws examined in this study was integrated care. For example, although the Care Act of 2014 did not mention rehabilitation explicitly, it does include provisions for integrating care and support with health services [29]. While the aim of the Health Act of 2009 is to improve the quality of health care and NHS services [30], rehabilitation is not explicitly addressed. The Health and Social Care Act passed in 2012 mandates the NHS Commissioning Board with duty to promote integration of health services provision [31], but does not address rehabilitation services directly. Although the initial National Health Service Act in 1966 did not mention rehabilitation, the later act in 2006 includes rehabilitation in its description of a hospital, stating that it is “any institution for the reception and treatment of persons during convalescence or persons requiring medical rehabilitation” [32].

Mixed results were found with respect to policies. While no specific rehabilitation strategies were found at the national level, a commissioning guidance for rehabilitation was published by NHS England. It outlines the scope of rehabilitation, components of high-quality rehabilitation, and how to compare rehabilitation services at all levels, including the national level [33]. The primary programmatic area where rehabilitation is acknowledged and discussed is cancer—the cancer strategy in England entitled “Achieving world-class cancer outcomes” addresses key calls to action in the WHO Rehabilitation 2030: Call for Action report. It discusses the need for a strong multidisciplinary rehabilitation workforce and promotes the role of allied health professionals in multidisciplinary teams. Similarly, the dementia strategy identifies the lack of access to rehabilitation care and addresses similar action areas to the cancer strategy, additionally highlighting the importance of intermediate care [34].

### 3.2. Netherlands

The Netherlands’ health system review document of 2016 describes the organization of the health care system of the Netherlands as a Bismarckian system centered around social insurance [35]. Recent significant health reforms have restructured the Dutch health system with the government taking on a more distant role in the health care sector. A major reform after 2006 shifted focus to introducing markets consisting of health insurance packages that are universal in nature, as well as provisioning and purchasing of health care. Thus, the organization of the health system in the Netherlands is decentralized, with many independent actors in the health sector adopting more responsibility.

The most prominent law detailing the structure and provision of health insurance is the Healthcare Insurance Act. Introduced in 2006, the act was the culmination of many years of policy making and legislation around health care access issues aiming to ensure a universal package of health services for all people in the Netherlands. Since its enactment, it was modified and the benefits package was subject to several changes due to various policy and economic considerations [36]. According to an amendment passed in 2015, specialist inpatient rehabilitation care is included in the benefit package of basic health insurance [35]. Physical therapy services, however, are not included in the statutory health insurance coverage package (with the exemption of those under 18 years old), and adults with mild impairments or chronic conditions who require physical therapy need to pay out-of-pocket for the first 12–20 sessions within a calendar year [35].

The Act on Access to Health Insurance, passed in 1998, is one of the few laws in the Netherlands that addresses financing of rehabilitation care. In Article 5, it is explained that health insurance includes reimbursing costs related to specialist medical care, which includes rehabilitation care [37]. 

While the Long-Term Care Act outlines provision of care for those requiring care on a daily basis or 24-hour home care due to physical or mental disabilities or impairment such as nursing care and personal care, it does not explicitly address the provision of rehabilitation services [38,39]. Similarly, the Social Support Act enacted in 2015 outlines various forms of support to persons with disabilities to increase self-independency and ability to maintain residence at home for as long as possible, but makes no mention of rehabilitation services [38].

Of all Dutch policies retrieved relevant to specified programmatic areas, few address rehabilitation services directly. Rehabilitation care is addressed in the action plan entitled “Program Longer At Home Working Together—Plan of Action 2018–2021”, which focuses on integration of rehabilitation in the health sector and inclusion of rehabilitation when developing a multidisciplinary workforce [14,40]. Specifically, this plan emphasizes the importance of geriatric rehabilitation care through incorporating supplementary specialist care into the Healthcare Insurance Act [40]. Certain policies address rehabilitation services indirectly. For example, the Netherlands’ national health action program for chronic pulmonary diseases identifies the importance of early rehabilitation [41], addresses key action areas to improve integration of rehabilitation into the health sector, and builds comprehensive service delivery models inclusive to rehabilitation care [14]. In addition, the national health policy document, ”Health Close to People”, mentions stroke rehabilitation as one type of care monitored by the Health Care Inspectorate [42]. Like the UK, the Netherlands’ policy on managing diabetes care does not address rehabilitation services.

### 3.3. Sweden

Sweden’s health care system is publicly financed and involves supplementary insurance coverage [43]. Unlike the Netherlands, the government plays a major role and all three levels of the Swedish government are involved [44]. Health care is recognized as a basic human right and UHC is provided to all legal residents [44]. Although primary care is at the forefront of services covered in health care delivery in Sweden [43], many other types of services are included in public coverage, including rehabilitation services [44]. Nearly all facets of health care services (e.g., inpatient and outpatient care) are financed and managed by the county councils, municipalities, and the government [43]. Responsibility for nursing and rehabilitation specifically falls under the responsibility of municipalities, while medical treatment is the responsibility of the county councils [44].

A law on financial coordination of rehabilitation services outlines ways to facilitate efficient use of resources [45], with the overall aim of improving ability to perform employment [45]. In addition, the law outlines the municipality’s responsibility to rehabilitation service provision alongside other services, such as nursing and physiotherapy [46]. An amendment to the Health and Medical Services Act in 2014 further outlines the right to rehabilitation services for residents within the jurisdiction of the county council, highlighting rehabilitation planning in collaboration with the specified individual [47]. However, amendments to the act made after 2014 did not address rehabilitation specifically. 

Despite coverage of rehabilitation services in Swedish legislation, no separate policies on rehabilitation specifically were found, although there is a strategy on vocational rehabilitation. Sweden’s national mental health strategy emphasizes the right to healthcare and rehabilitation [48].

### 3.4. Germany

The German health care system is divided into inpatient care, outpatient care, and rehabilitation care [49]. Rehabilitation facilities in particular are responsible for provision of treatments to restore functioning and independence, such as physiotherapy [49].

The legal basis of service delivery outlined in the Social Code Book IX includes reference to rehabilitation services [50], with provision to ensuring rights of disabled persons [50]. Coordination of services is also highlighted in the law, where person-centered care is emphasized and rehabilitation agencies demonstrate their responsibility in providing care through efforts to be effectively integrated with individuals seeking care [50]. In general, the German law is extensive and covers many areas subject to rehabilitation care in health systems, including services, delivery, coordination of care, and financing by means of insurance schemes. 

The National Action Plan 2.0 of the Federal Government to the CRPD is a cross-sectoral action plan on disability that directly addresses rehabilitation. Although a large emphasis is placed on vocational rehabilitation and restoring employability, key actions focus on rehabilitation services to enable and promote social participation for all people in all areas of life [51]. In terms of inclusion of persons with disabilities, the National Action Plan 2.0 aligns with the WHO’s Global Disability Action Plan 2014–2021 Better Health for All People with Disabilities [52]; overlapping actions include strengthening the provision of rehabilitation in the community for persons with disability and extending the provision of assistive technology and assistance and support services for persons with disability. Furthermore, rehabilitation measures include support for disabled persons and their families. In addition, there is emphasis on rehabilitation services for the prevention of impairments leading to disability [53].

### 3.5. Italy

Like Sweden and the UK, basic primary health coverage in Italy is provided for by national health services [54]. What is unique to Italy is that its national health service is universally covered and regionally based [55], with the fundamental principles of health care in the NHS laid out at the national level, and the regional level of government being responsible for the delivery of health services via local health authorities [55].

Many Italian laws discuss rehabilitation services. For example, Law No. 104 passed in 1992 directly addressed rehabilitation care in achieving optimal functioning for persons with disabilities [56], and Article 7 focuses on rehabilitation care, creating programs to allow disabled persons to be fully integrated in all aspects of life [56]. The article further states that the NHS in Italy is responsible for ensuring early rehabilitation treatment and care, as well as provisions for assistive devices and technology for disabled persons [56]. This information suggests that this law recognizes rehabilitation as a basic human right. Law No. 328, passed in 2000, indirectly addresses rehabilitation care in relation to individual projects for disabled persons. Article 14 includes rehabilitation care and treatment provisioned for by the NHS when describing the needs and services for individual projects to integrate disabled persons into society [57]. 

In Italy, there is a commitment to rehabilitation care, as represented by the Rehabilitation National Plan: An Italian Act. This policy document discusses strategies to appropriately implement rehabilitation services as outlined in the legislation. For instance, individualized rehabilitation plans centered around the needs of an individual are emphasized [58]. In terms of other key programmatic areas relevant to health, rehabilitation services are both indirectly and directly addressed. 

## 4. Discussion

In this study we were especially interested in examining how approaches to ensuring and expanding access to rehabilitation care differ among countries, and what lessons can be learned from such an analysis. Our effort is not to identify a best case or a single effective or recommendable approach to rehabilitation policy development, as there are potentially several equally effective ways to position rehabilitation in a country’s health policy. Approaches to integrating rehabilitation in UHC frameworks reflect the particularities of a country’s health system, the values and evidence on rehabilitation benefits considered in decision making processes, the historical development of the country social welfare system, and the availability of resources and infrastructure. In the discussion below we consider the situation regarding rehabilitation services in each country, considering WHO’s Rehabilitation Call for Action.

### 4.1. Positioning of Rehabilitation in National Legislation and Policies

The results summarized in Table A1 and Table A2 regarding the provision of rehabilitation services in national health laws and policies in the specified countries show several commonalities and differences.

Nearly all countries (apart from the Netherlands) address rehabilitation services through health laws, which suggest there is a strong legal basis for rehabilitation service provision. However, while rehabilitation is considered a key component of health service delivery, the focus of State regulation regarding rehabilitation varies. For example, Sweden’s publicly financed health care system showed greater emphasis on regulating important aspects of rehabilitation services, especially financing. The strong legal basis for rehabilitation in Sweden can be explained by the historical context surrounding its legislative framework, as there was a large need to rehabilitate persons after the Second World War [59].

Not all UHC-centric countries explicitly call out provision of rehabilitation care in health laws. One prime example is the UK, which did not address rehabilitation care specifically, but rather placed emphasis on integrated care in its legislative framework, as shown by laws such as the Care Act of 2014 and the Health and Social Care Act passed in 2012. By contrast, Germany includes rehabilitation provision and delivery in health laws despite its system being driven by statutory health insurance. Prime examples include the Federal Act on Participation, which emphasizes the importance of rehabilitation services with respect to social participation [60], highlighting Germany’s commitment to a legislative framework inclusive towards persons with disabilities and their need for access to appropriate rehabilitation service provision and delivery. 

It should be noted, however, that absence of state policy in an area does not necessarily mean the absence of regulation. Increasingly, health care professionals have taken initiative to regulate aspects of care delivery with the aim to improve patient experiences of care and quality of care through standardization of care processes, implement evidence-based guidelines, as well as continue in professional development and training. For example, the European Union of Medical Specialist Physical Rehabilitation and Medicine (PRM) board published guidelines for the treatment of people with various disorders that can be adopted by national societies of PRM physicians and greatly improve quality and effectiveness of care [61,62,63]. Other organizations also published standards for the delivery of rehabilitation in various settings [64]. In the UK, more recently the Chartered Society of Physiotherapy (CSP) published guidelines for the treatment of people affected by COVID and the provision of rehabilitation in the community for those who have recovered from COVID [65]. These are important initiatives that have the potential to expand access to rehabilitation and improve quality of care for a wide range of beneficiaries by stimulating changes in health care processes, national policies, and legislation. 

Regarding persons with disabilities, countries examined in this study have not explicitly addressed the right to rehabilitation in national legislation, with the exception of Germany and the Netherlands. As indicated by the results tabulated in Table A1 and Table A2, legislation regarding disability rights mostly concerns the right to equality and nondiscrimination, as well as the right for disability policy or other issues. Contrarily, there is high prioritization of rehabilitation rights for persons with disabilities in Germany and the Netherlands’ cross-sectoral laws. This may be explained by historical and cultural factors, i.e., Germany’s move away from a system of welfare to a participatory law encouraging full participation of disabled persons at the forefront of German society [66]; social reforms in the Netherlands also triggered new goals for the State to encourage participation in the workforce [67]. In the United Kingdom, Sweden, and Italy, a lack of emphasis on promoting social participation of persons with disabilities may explain the lack of prioritization of rehabilitation care for disabled persons in their cross-sectoral legislations. Good governance requires that countries recognize the right to rehabilitation for persons with disabilities in national legislation [1].

The discrepancy observed in the provision of rehabilitation services for persons with disabilities between cross-sectoral and sectoral health laws can also be seen in health policies, as shown in Table A2, again with the exception of Germany and the Netherlands. In fact, the proportion of cross-sectoral policies in comparison to sectoral health policies issued by the Ministry of Health—regardless of rehabilitation service provision—is low. Furthermore, within sectoral policies and action plans, gaps were found in health policy development regarding rehabilitation services for disabled persons. Germany’s more comprehensive national response to rehabilitation needs for disabled persons may be explained by the nation having the highest percentage of self-reported chronic morbidity (42.3%), as shown in Table 1. 

While the legal basis for the right to access and benefit from rehabilitation is present in the countries’ legislations, government commitment to implement rehabilitation care in national strategies and action plans is lacking. Interestingly, the primary country where national response to rehabilitation care needs is prominent is Italy, which, despite its fragmented health system, is committed to implementing rehabilitation provision in its policies. 

Key trends were also observed in health programmatic areas across the countries. For instance, provisions regarding rehabilitation care was largely absent from diabetes action plans, which focused on prevention, when compared with comprehensive rehabilitation plans in cancer care strategies. Finally, key action areas addressed in the WHO Rehabilitation 2030: Call For Action report, such as creating strong leadership and political support for rehabilitation at all levels, were not addressed in many action plans retrieved [14].

### 4.2. Study Limitations

This study had several limitations. A limited time frame in which the study was conducted did not allow for a fully comprehensive search of the grey literature. The study focused only on laws and policies at the federal or national level and legal instruments governing the provision and financing of rehabilitation at the regional level are not presented. Despite the use of online translation software, contextual content may have been lost to reviewers’ limited knowledge of Dutch, Swedish, German, and Italian. Furthermore, researcher bias may have contributed as a limitation in this study. We have deliberately chosen to focus on five countries assuming that these share similar characteristics and health system challenges, despite their diversity in approaches to developing and implementing health policies. The small sample of countries and the purely descriptive nature of the study does not allow for generalization on approaches to rehabilitation policy development. Finally, due to the descriptive nature of this study, we were not able to assess or infer the degree of implementation of rehabilitation legislation and policies. Although the existence of national legislation and action plans are important indicators of a country’s commitment to strengthening rehabilitation [68,69], particularly at the level of governance, the absence of such instruments is not sufficient to make any claims regarding access to rehabilitation. Future research is needed to assess the level of accessibility to rehabilitation services considering the linkage between access to health services and health policy parameters discussed in this review, especially concerning insurance coverage and benefits package design.

## 5. Conclusions

The goal of this research study was to describe health laws and policies and assess the degree of prioritization of rehabilitation care in the legislative and policy landscape in five European countries. Historical, cultural, demographic, and population aging factors may explain the commonalities and differences found between all health laws and policies among the specified countries. 

Policy making and development is a cornerstone of health care implementation. Policy analysis as a research tool can provide important insights towards understanding regional health care objectives and achieving integrated care goals. At present, rehabilitation-specific policies and legislations are not featured strongly in health policy and systems research. The findings generated from the synthesis have important implications for rehabilitation policy development in Europe, emphasizing the need for States’ governments to support rehabilitation care access through their policies and action plans in Europe. This study is especially significant because it is the first attempt to map the terrain of rehabilitation services policy development in Europe, and highlights the need for further research in this field to obtain a more comprehensive picture of the degree of implementation of rehabilitation services at both the policy and legal level.

Overall, the findings show that despite recognition of the importance of rehabilitation in national legislation, governments’ commitment to strengthening rehabilitation within and across areas of health programming is weak. The lack of strategic plans and rehabilitation specific policy frameworks show that rehabilitation remains an invisible facet of national health policy development. There is also a need for more systematic and accurate evaluation of government-level interventions as well as for a stronger focus on strategic planning rather than improvements in service delivery and outcomes alone.

The results obtained from conducting this narrative review is a pivotal starting point in the monitoring and evaluation of access to rehabilitation care and provide a basis for future appraisals of the situation regarding rehabilitation services in Europe. 

## Figures and Tables

**Figure 1 ijerph-17-04363-f001:**
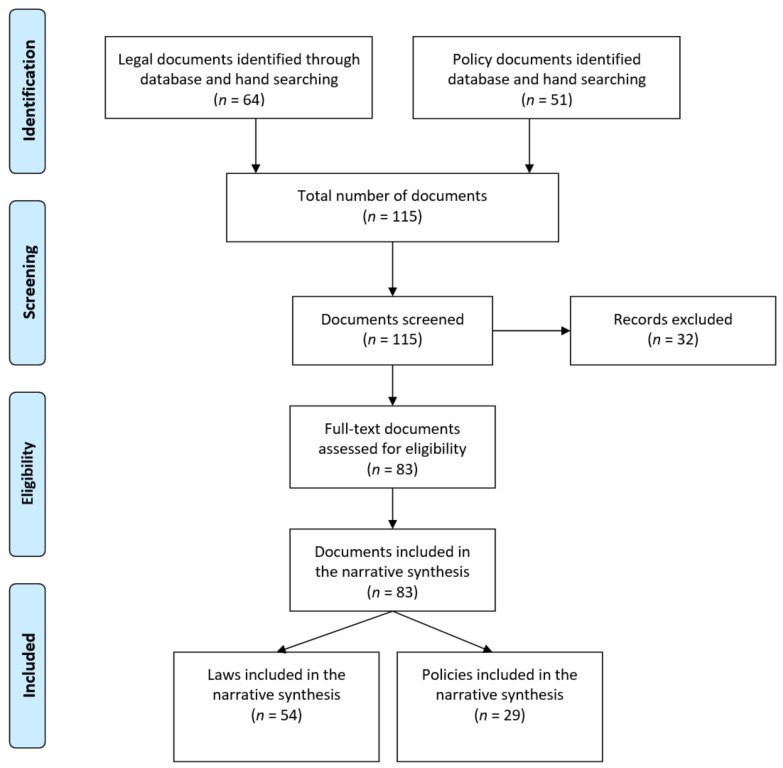
Document identification and selection flowchart [25].

**Figure 2 ijerph-17-04363-f002:**
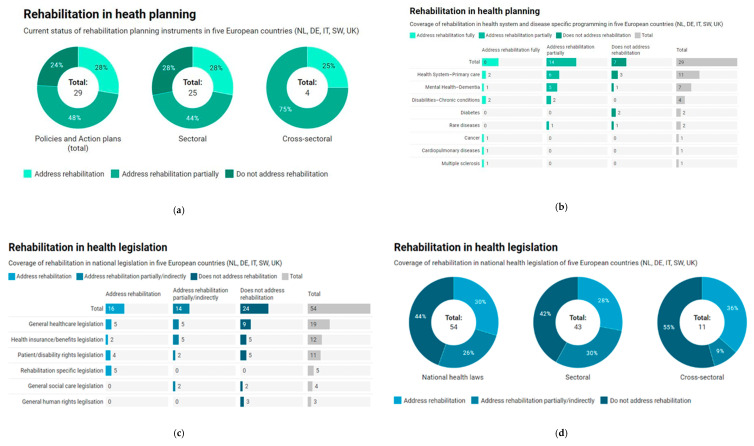
Coverage of rehabilitation in health law and planning instruments in five European countries (NL, DE, IT, SW, UK): (**a**) current status of rehabilitation in health planning instruments, (**b**) coverage of rehabilitation in health systems and disease specific programming instruments, (**c**) coverage of rehabilitation in national health legislation, and (**d**) coverage of rehabilitation in national legislation.

**Table 1 ijerph-17-04363-t001:** Basic information on countries included in the study.

	United Kingdom	Sweden	Germany	Italy	Netherlands
Welfare model type	Anglo-Saxon/Liberal	Nordic/Social Democratic	Continental/ Conservative	Mediterranean	Continental/ Conservative
Healthcare system model	Beveridge	Beveridge	Bismarck	Beveridge/Mixed	Bismark/Mixed
Subregion	Northern/ Nordic	Northern/ Nordic	Continental Europe	Southern Europe	Continental Europe
Characteristics	National Health Service System funded through general taxation	National health care system funded through general tax revenues; regulation, supervision, and some funding through national government; responsibility for most financing/purchasing/provision devolved to county councils	Statutory health insurance system; system funded through employer/ employee earmarked payroll tax and general taxation	National health care system funded by national earmarked corporate and value-added taxes as well as general/regional taxation; funding and minimum benefit package defined by national government; planning and provision by regions	Statutory health insurance system, with universally mandated private insurance; funded through earmarked payroll tax; community-rated insurance premiums; general tax revenue
	Extensive network (mainly private) of primary care providers	Mixed primary care system (40% private, 60% public)	Private primary care system	Private primary care system	Private primary care system
	No cap on cost sharing Drug cost-sharing exemptions	Caps on cost sharing (Annual maximum for outpatient visits is SEK 1,150 (USD 125); for drugs, SEK 2,250 (USD 246) for adults); Some cost sharing exemptions	Cap on cost sharing (2% of household income, 1% of income for chronically ill); children and adolescents <18 years of age exempt	No cap on cost sharing; exemptions for low-income older people/children, pregnant women, chronic conditions/disabilities, rare diseases	No cap on cost sharing; annual deductible of 385 Euros covers most cost sharing; general practitioner care and children exempt from cost-sharing;
**Key Indicators (2016)**					
Healthy Life Years at age 65 (men, women)	10.4	15.1	11.5	10.4	10.3
	11.1	16.6	12.4	10.1	9.9
Population aged 65 and more (%)	17.9	19.8	21.1	22	18.2
Self-reported chronic morbidity(%) ^a^	36	37.6	42.3	15.2	33
Rehabilitation expenditure per capita (PPS)	*n*/a	*n*/a	117.8	*n*/a	168.3
Rehabilitation beds/1000 population	*n*/a	*n*/a	2.01	0.41	0.11
Practicing physiotherapists/100,000 population	44.49 (2014)	129.2 (2012)	207.8 (2013)	93.88 (2013)	192.4 (2013)

^a^ Proportion of people reporting any long-standing chronic illness or longstanding health problem. Source: The Commonwealth Fund (International Profiles of Health Care Systems 2018), European Commission (European Core Health Indicators—ECHI), OECD, and WHO European Health Information Gateway.

**Table 2 ijerph-17-04363-t002:** Inclusion and exclusion criteria for retrieval of health legislation and health policies.

	Legislations	Policies
Criteria	Inclusion	Exclusion	Inclusion	Exclusion
Type	Enacted legislation detailing financial and institutional arrangements on health service coverage/delivery	Laws on human research	Action plans, strategies, and policies addressing the following target groups:	Action plans focusing only on prevention of noncommunicable diseases
-People with noncommunicable diseases: cancer, stroke, diabetes, cardiovascular disease, chronic obstructive pulmonary disease-Older persons (i.e., aging)-Persons with mental health and neurodegenerative disorders (e.g., dementia, Parkinson’s disease, multiple sclerosis)-People with disabilities and/or chronic conditions	
Laws on data protection including regulations regarding the protection of privacy and confidentiality of health information		Action plans, strategies, and policies addressing other aspects of health service delivery such as e-health/digital health
	Action plans, strategies and policies addressing the following programmatic areas within the health system, including:
Laws and regulations on health professional practice and education	-Human resources-Health financing-Service user empowerment-Health information systems governance

Environmental health laws	Intersectoral action plans on disability (i.e., national strategies for the inclusion/integration of disabled persons)

Food laws	Sectoral or multisectoral action plans on rehabilitation

Submitted bills, upcoming bills, law proposals, national constitutions
Timeframe	1980–2018	Pre-1980	Action plans, strategies and policies that are currently under implementation or were implemented within the last three years, i.e., implemented by 2015.	Action plans, strategies, and policies completed preceding 2015
Level of implementation	National	Municipal, Regional	Action plans, strategies and policies at the national or federal level	Action plans, strategies and policies at regional, municipal, cantonal, and provincial levels
Languages	English, German, Swedish, Italian, Dutch	All languages not specified in inclusion criteria	English, German, Swedish, Italian, Dutch	All languages not specified in inclusion criteria
Countries	United Kingdom, Italy, Netherlands, Sweden, Germany	All countries not specified in inclusion criteria	United Kingdom, Italy, Netherlands, Sweden, Germany	All countries not specified in inclusion criteria

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
