# Peer review of "Legal and Regulatory Approaches to Rehabilitation Planning: A Concise Overview of Current Laws and Policies Addressing Access to Rehabilitation in Five European Countries"

_ijerph, 2020, doi:10.3390/ijerph17124363_

Round 1

Reviewer 1 Report

Dear authors,

Thank you very much for the thoughtful manuscript in an important field.

In my opinion, gray literature is essential information for this type of study. Today, the option of using text analysis techniques for exploring hidden information from a large-text corpus is very suitable. Without the gray literature, the word: regulatory in the title of the manuscript could be misleading. 

Besides, for me, the length of the manuscript was too long. I suggest reducing the specific writing about each country. 

Minor comments:

Numbers in the abstract.

Please specify your Key-words

Tables: please justify the words to left

Table 4: please add a total row at the end of the table, presenting the numbers of the issues (color) in each column.

Author Response

We appreciate your insightful comments regarding our manuscript submission, and would like to thank you for taking the time to comment on our manuscript and providing meaningful insight.

No

Reviewer comment

Authors response (please see track changes in Manuscript)

1

In my opinion, gray literature is essential information for this type of study. Today, the option of using text analysis techniques for exploring hidden information from a large-text corpus is very suitable. Without the gray literature, the word: regulatory in the title of the manuscript could be misleading.

We have characterized the type of literature as gray literature in the methods section, see lines 8-9 (page 6).

2

the length of the manuscript was too long. I suggest reducing the specific writing about each country.

We have shortened the text within each country subsection – please revisit results section (lines 202-338; pages 15-17).

3

Numbers in the abstract.

We were unable to address this comment as numbers in the abstract are required by the journal.

4

Please specify your Key-words.

Keywords are specified after the abstract in line with Medical Subjects Headings terms (MeSH).

5

Tables: please justify the words to left

Words in tables have been left justified and formatted according to journal requirements– please see pgs 4-5, 9-10, 21-30,

6

Table 4: please add a total row at the end of the table, presenting the numbers of the issues (color) in each column.

Row was added to bottom of table 4 to highlight counts of each category. We did the same for Table 3 for consistency. The tables have been removed from the manuscript on request of another reviewer and appear as an online appendix.

Reviewer 2 Report

This paper represents a really interesting piece of research, as well as a starting point for future appraisals of the situation regarding rehabilitation service and policy development in European Countries.

The paper is clear and well structured, also thanks to the tables and figures attached and the additional ones provided online. I only suggest to consider revising the layout of table 1 (p. 4) and table 2 (p. 8) to render them more readable (maybe changing the text alignment from the centre to left or justified, and dividing columns and rows with visible lines). 

Finally, I just signal a misprint at line 66 (compl\ications) and a possible shift of the caption of figure 2 at line 125.

Author Response

We appreciate your insightful comments regarding our manuscript submission, and would like to thank you for taking the time to comment on our manuscript and providing meaningful insight.

No

Reviewer comment

Authors response (please see track changes in the document)

1

suggest to consider revising the layout of table 1 (p. 4) and table 2 (p. 8) to render them more readable (maybe changing the text alignment from the centre to left or justified, and dividing columns and rows with visible lines). 

Tables have been justified left, and reformatted to enhance clarity and follow journal guidelines. Please see pgs 4 and 9-10.

2

misprint at line 66 (compl\ications)

The misprint has been fixed – please see line 67.

3

possible shift of the caption of figure 2 at line 125

The indent has been removed – please see line 152 (pg. 13).

Reviewer 3 Report

Thank you for the opportunity to review this manuscript. I believe it has the opportunity to be quite a seminar paper in the current evolution of rehabilitation in health systems. The paper comes at a time when the rehabilitation sector is enjoying new voice and power to shape an ambitious agenda. The challenge of operationalising broad calls for 'better policy, better integration' and so on remains reasonable guidance and a body of work to inform new efforts. As these researchers have shown, rehabilitation 'policy' is a 'cloudy mesh' of legislation, policies that have taken generations to develop. In light of calls to deliver policy reform, this work provides a reality check about what it is going to take to deliver policy reform, and an extremely useful mapping of current legislation. 

The authors argued that this work is urgent because the needs are growing, but services are not - and that new global calls renew focus on rehabilitation in health systems. They systematically review relevant policy and legislation, organise it according to its level of consideration of rehabilitation, and synthesise the state of the art in a very useful comparison of some illustrative countries. 

I have no hesitation in commending this paper to the editors for publication. It is a superb starting point for the next phase of evolution of rehabilitation in health systems. 

I provide some line-by-line editorial and substantive comments below. 

15 - I suggest the prevalence of functional impairments is a better argument for rehabilitation than prevalence of disability. Framing rehabilitation as a mostly (or only) response to disability is problematic both for the rehabilitation sector and the disability movement.

19 - '..is increasingly acknowledged', not 'is being increasing acknowledged'. Either way the sentence is quite passive and could be stronger.

21 - is it 'unclear' or under-researched? Or rather 'has not been synthesised or similar'.

  1. Please check your interpretation of this seminal reference. Is it 2-4% of 15% or 2-4% of the population overall who are estimated to experience significant difficulties? Regardless, why is the prevalence of disability the important statistic here? Shouldn't it be all-cause need for rehabilitation?
  2. Here again, and throughout, the argument is situated as a disability argument, rather than responsiveness of health systems. I think this needs a re-think: is the argument for the work a response to disability rights or the fit and prioritization of services for the population. This distinction forms the basis of different policy implications and interpretations and so needs to be abundantly clear at the outset of this work.
  3. I suggest it's important not only to talk about the increasing numbers of people with disability, but the increased prevalence of co-morbidities as well. That is the problem of rehabilitation policy lag is not just about the *amount* of rehabilitation, but its *quality and sophistication*, too. For consideration.
  4. Check typo 'complications'
  5. Characterising the avoidance of complications from primary health conditions seems to be a small subset of pathways to reduced healthcare costs. Can a fuller argument for potential efficiencies be presented briefly here? Further, it seems at odds with the definition of rehabilitation provided earlier in the paragraph.
  6. This line sums up what I think is the weakness of this section. Why does rehabilitation need to be characterised variously as a part of responsive health services overall, or specifically for people with disabilities? There is no disputing that people with disabilities have specific rights to rehabilitation, that many (but not all) can derive a disproportionate benefit from quality rehabilitation and that perversely, they can be precluded from accessing the right care. However, the aim here is to describe rehabilitation policy and law. Intersections with disabilities are one important component, but here we revert back and forth between situating rehabilitation as a health response, a disability response, and so on. It's confusing and nothing is lost with a much simpler comment about people/persons with disabilities being a very important population whose particular rights should inform analysis of rehabilitation policy.
  7. Aren't there 4 questions here? 1, how do national laws govern rehab, 2, what aspects are subject to State regulation, 3, how is rehabilitation addressed in health policies, 4, what are common/divergent approaches to policy development.

Would it be clearer to present 4 clear questions rather than 2 compound questions?

  1. is it ‘have chosen to’ or simply ‘have’?
  2. The table is hard to read. Can lines be added?

**Line Numbering resets

  1. What ‘WHO terms’ are being referred to here? Was rehabilitation medicine or nursing a term? Rehabilitation engineering, vision rehabilitation? There are complex, overlapping taxonomies that materially affect your methods and interpretation and this section should be very clear about a) how you decided on your terms, b) precisely what the terms were and c) why this matters to your interpretations.
  2. The table is very hard to make sense of. I suggest top align of cell, indents, etc.

The inclusion criteria for policies in this table speaks to my earlier point about the relative emphasis on disability in the introduction. Persons with disabilities are one group here of 4, explored. Why then the emphasis on disability in the background?

Consider whether ‘persons’ or ‘people’ is the right term. The convention of ‘persons with disabilities’ being used when discussing persons with disabilities as right-holders seems to be reasonable but I don’t believe it should carry over to ‘people’ with NCDs, etc. For consideration.

  1. You use the term ‘allied health’ here for the first time. I suggest you define it. I suspect you’re referring to the subset of rehabilitation professionals who might be called allied health professionals – are all rehabilitation professionals ‘allied health’?
  2. Ensure caption and image are not separated
  3. I like this table. However, it is very long. The colours might not work well in print form. Could a number or even the words ‘direct/indirect/none’ be used?

It is concerning that the offender rehabilitation act appears, here: why? Should rehabilitation of criminal offenders be an exclusion criterion? A look at that particular act confirms there is nothing relevant to rehabilitation by your definition.

What is the value of the ‘cross sectoral’ law columns? Whether or not they are cross sectoral, they refer to rehabilitation. This occupies too much space for the point being conveyed.

I suggest the table be provided as an appendix to the paper at the editors’ discretion. Figure 2 conveys the more interesting information and is sufficient to answer the main aims.

  1. This line makes no sense to me. Nearly have ‘does’ is about the same as ‘nearly have do not’. Is this simply trying to say that about half of the policies at least partly addressed rehabilitation – the remaining did not mention rehabilitation at all.

This paragraph seems to be longer than necessary. I think you’re just trying to say that there are more references to rehabilitation in policy than in legislation: about a half and a third respectively addressed rehabilitation.

  1. This is the most essential and important paragraph for me. It is superb. I suggest though that ‘coverage’ is not the right term. Is it ‘consideration of rehabilitation’ or similar?

158 Check ‘rehabilitation in health planning’ figure column headings. Replace ‘full address/partial address/no address with ‘fully addressed/partially addressed/not addressed.

167 ‘The UK’s health care system is primarily publicly financed by…’ (i.e. delete ‘one that’)

  1. What do you mean ‘divides the shift in focus’? Please rephrase.

This paragraph seems to summarise citation 114 (Annel 2014) which at would appear from the title to concern Sweden. If the citation is to be used like this, refer to it by name, and directly cite their explanation of the Dutch scenario, rather than re-phrasing it sentence-wise and citing it repeatedly.

  1. Replace ‘support in the rehabilitation space’ with ‘rehabilitation services’

It seems to me the sub-heading here really mostly tackles half of the second aim ‘what are differences between countries’. I can intuit from the discussion important findings about the other parts of your aims: how is rehabilitation reflected, where should it be, but isn’t, and so on. Maybe it is just the wrong sub-heading. This section does progress very important, useful answers to your main questions, so I suggest being much clearer about that.

380 – perhaps a discussion point, perhaps a limitation – but you need to remark on the implementation or realization of lofty policy goals. That is, to what extent are policies enacted, and laws upheld? Wouldn’t we be better off in a country with good access, but poor regulation?

Further, I think you need to remark on self-regulation. To what extent do professions govern themselves? Are there interprofessional networks, perhaps for disease-specific groups of professionals, for instance? I.e, is the absence of state policy absence of regulation?

  1. I am not convinced you have quantified ‘prioritization’ or rehabilitation. That is anyway less important than what you have done: provided a useful taxonomy of existing laws and illustrated the broad point that a) rehabilitation cuts across multiple policy dimensions, b) is variable between countries c) depends on history, context, and d) (my addition) might anyway not be the main driver of what is actually delivered in practice.

I also suggest that you have not contrasted rehabilitation with ‘other health strategies’ (line 401). Again, that does not necessarily detract from the work, but I don’t think you can draw that conclusion.

I suggest the conclusion should provide at least one or two sentences about your findings. Some variation of ‘policy varies between country, rehabilitation is reflected variously across multiple policies, isn’t where we would expect it’ etc.

Author Response

We appreciate your insightful comments regarding our manuscript submission, and would like to thank you for taking the time to comment on our manuscript and providing meaningful insight.
